# Threat Modeling for Communication Security of IoT-Enabled Digital Logistics

**DOI:** 10.3390/s23239500

**Published:** 2023-11-29

**Authors:** Aisha Kanwal Junejo, Michael Breza, Julie A. McCann

**Affiliations:** 1School of Computer Science and Mathematics, Keele University, Newcastle ST5 5BG, UK; a.junejo@keele.ac.uk; 2Adaptive Emergent Systems Engineering, Imperial College London, London SW7 2BX, UK; michael.breza04@imperial.ac.uk

**Keywords:** digital logistics, tracking, security, threat model, attacks, vulnerabilities

## Abstract

The modernization of logistics through the use of Wireless Sensor Network (WSN) Internet of Things (IoT) devices promises great efficiencies. Sensor devices can provide real-time or near real-time condition monitoring and location tracking of assets during the shipping process, helping to detect delays, prevent loss, and stop fraud. However, the integration of low-cost WSN/IoT systems into a pre-existing industry should first consider security within the context of the application environment. In the case of logistics, the sensors are mobile, unreachable during the deployment, and accessible in potentially uncontrolled environments. The risks to the sensors include physical damage, either malicious/intentional or unintentional due to accident or the environment, or physical attack on a sensor, or remote communication attack. The easiest attack against any sensor is against its communication. The use of IoT sensors for logistics involves the deployment conditions of mobility, inaccesibility, and uncontrolled environments. Any threat analysis needs to take these factors into consideration. This paper presents a threat model focused on an IoT-enabled asset tracking/monitoring system for smart logistics. A review of the current literature shows that no current IoT threat model highlights logistics-specific IoT security threats for the shipping of critical assets. A general tracking/monitoring system architecture is presented that describes the roles of the components. A logistics-specific threat model that considers the operational challenges of sensors used in logistics, both malicious and non-malicious threats, is then given. The threat model categorizes each threat and suggests a potential countermeasure.

## 1. Introduction

Digital logistics will transform supply chain and delivery operations by providing real-time asset tracking and monitoring through the use of small-sensor-based systems such as Wireless Sensor Networks (WSN), wireless embedded sensor devices, edge computing and low-power long-range wireless communication technologies, sometimes collectively called the Internet of Things (IoT) [1]. Digital logistics reduces risk in a cost-effective way and allows for better supply chain management.

Many major companies have incorporated the use of IoT devices into their shipping and supply chain. Ship owner and operator Maersk [2], tire manufacturer Michelin [3], and major international global courier DHL [4] are progressively digitizing their supply chains through the use of wireless embedded sensor devices. Maersk uses IoT devices on containers and the Azure IoT Hub as a cloud-based backend solution to monitor shipments [2]. Michelin uses SigFox IoT devices to manage its intercontinental sea freight flows by monitoring the physical movement (shipment loading and unloading) and condition (temperature, humidity and shocks) of the its shipped assets. Global courier DHL employs Sigfox-enabled embedded sensors to track its deliveries [4].

IoT-enabled tracking devices in shipped assets also can help to combat instances of fraud, a common problem in the shipping industry. A recent shipment of 1104 containers of the metal nickel arrived at the purchaser containing no nickel. The fraud cost the commodity trader over half a billion U.S. dollars [5].

This paper stems from a project that had the remit to organize a shipment of goods from the UK to Singapore using electronic trade documents in the form of an Electronic Bill of Lading (E-BL) which persisted on a distributed ledger (DLT) [6].

Our task was to provide the location and condition data of the goods during shipping via IoT sensors, to provide a secure physical-to-digital link between the shipped goods, and the E-BL stored on the DLT.

We found that there were no threat models that covered all of the threats, both malicious and non-malicious for such as system. It is important that a threat model for an asset condition monitoring and tracking system covers as many threats as possible, and does not introduce new security flaws or provide new opportunities for fraud.

The clearest and most significant vulnerabilities that arise with IoT technologies in logistics are rooted in machine-to-machine (M2M) communication, either unicast or broadcast [7]. Wireless communication systems are vital to the operation of IoT systems, used to synchronize and manage the sensors and enable user access to condition information and location. Wireless communication systems are particularly vulnerable to security attacks such as radio jamming, eavesdropping, man-in-the-middle (MitM), spoofing, replay, and malicious code injection. All of these attacks can be carried out without physical access to the IoT devices. Physical access enables further communication-based attacks such as device forging, cloning, impersonation, unauthorized tag reading, tag modification, and malicious code injection. Communication-based attacks violate the *confidentiality*, *integrity*, and *authenticity* security properties of the system.

In this work, we present the threat model that we used for our IoT logistics deployment and a reference system architecture for the use of IoT devices in digital logistics. Although we focus our discussion on our reference architecture, our threat model is general to any logistics application that uses IoT devices and their communication. We follow with a detailed survey of the cyber threats and attacks affecting our reference system architecture. We organize the threats in our survey into a threat model that specifically addresses the vulnerabilities of IoT systems used for digital logistics. We then present an approach to mitigate the presented risks.

To the best of our knowledge, this is the first study to investigate the security of asset tracking systems. We present a detailed review of state-of-the-art asset tracking systems to understand their tracking limitations when transported via different modes (road, sea, and air) all over the world. The threats and vulnerabilities are studied for a subset of the five layers of the Open Systems Intercommunication (OSI) model (or a superset of the TCP/IP communication model) typical of embedded system communication [8].

The rest of this paper is organized as follows. A background of current IoT/WSN threat models is presented and discussed in Section 3. Security issues specific to IoT/WSN systems used in logistics are also presented in this section. The system architecture and threat model are described in Section 4. The threat descriptions and a discussion of potential approaches to mitigate the presented risks are discussed in Section 5. Lastly, the conclusion and future research directions are discussed in Section 6.

## 2. Motivation

The scenario application was the shipment of goods from the UK to Singapore using electronic trade documents in the form of an Electronic Bill of Lading (E-BL) that persisted on a distributed ledger (DLT) [6]. Our task was to provide location and condition data of the goods during shipping via the IoT sensors and a physical-to-digital link between the shipped goods and the E-BL stored on the DLT.

We first describe the system architecture and the deployment, and then the logistics-specific considerations for the design of the system and its security.

### 2.1. System Architecture and Deployment

The IoT sensor system was a network of small, low-cost sensor tags, and a larger gateway device to collect the sensor tag data and send a summary of it to a server hosting the user-facing application.

Four sensors were placed at different locations on the shipping pallet and a gateway was placed at the top of sets of boxes; see Figure 1. The sensor tags monitored temperature, humidity, and acceleration. The sensed data were sent to the gateway which relayed it to a cloud backend. Sensors overheard the sensor data of their neighbors and compared it to their own to determine a level of trust that a sensor tag has for its neighbors; that is, they should be seeing similar measurements. Further details about the hardware used in this deployment are given in Section 4.

The IoT devices, four sensors and a gateway were commissioned with a secret key used for communication encryption and separate keys for authentication. All communication used Poly20ChaCha1305 encryption and formed a network, synchronized with one another. They sampled condition data (temperature, humidity and IMU) and sent that to a gateway periodically. The gateway synchronized the sensors and collected data from the sensors. The gateway encrypted and stored the data to local storage and sent the data to a cloud backend via LTE. The DLT periodically pulled data from the cloud backend, checked the authentication keys, and added the data to the DLT to accompany the E-BL. The cloud backend also provided cell-level localization to the data which were included with the sensor data. The system successfully provided condition and location data for a pallet of goods shipped from the UK to Singapore.

### 2.2. Logistics-Specific Design and Security Considerations

The purpose of the sensor system was to provide a secure physical link between the shipped goods and the E-BL stored on the DLT. We had to consider the security and reliability of the system while it was a part of an international shipment.

We found that three main aspects of the logistics operating environment affected the design of the sensor systems and its security.

No access to the sensor nodes; they have to work and continue to work with no physical access.Mobility causes periodic blind spots from the gateway to the backend.Uncontrolled access. Ports and shipping yards are large spaces which are hard to physically secure. A container or its contents may be tampered with at many points during the shipping process.

The final point is very important. Access can be gained to the devices or data without requiring physical contact. It is conceivable that the pallet sensing system could be attacked by someone outside of the ship, port, or shipping yard.

However, communication is not the only threat. The sensors could be non-maliciously damaged during the shipping process by a pallet lift, be exposed to a high-humidity environment that causes sensor damage, or suffer excess shock during transport. The fact that we had no physical access to the IoT system required us to consider all of these eventualities in the design of our system.

It is important to note at this point that the core property that we are designing for is reliability of the system. With this in mind, our threat model contains both malicious and non-malicious threats; all can degrade the reliability of the sensing system and so all must be included in the design. Another interesting point is that it may be difficult to discern between malicious and non-malicious events during shipping. We leave further discussion of detection accuracy to other work.

The logistics application contained enough unique aspects that we found that general IoT threat models were insufficient. In all cases, they focused only on malicious threats. As noted above, that focus is too narrow for the design of a production system. Other applications exist in operational environments that are easier, and require less strict design time planning. These aspects led us to develop a production-ready threat model that we could use for the design of our logistics sensing system.

## 3. Threat Considerations Specific to the Use of IoT in Logistics

### 3.1. Iot Devices in Logistics

Recent work discusses the use of IoT devices in logistics. Andreas [9] underlines the new trends and technologies, such as IoT, sensor devices, distributed ledgers, autonomous vehicles, and long-range (LoRa) communication, that are reshaping supply chains and asset tracking systems. Grzybowska et al. [10] provide a comprehensive guide to sustainable logistics and Industry 4.0 and cover methods, models and case studies that may assist in product automation. They also highlight the new opportunities and challenges in automating complex production processes and the tight coupling among them.

### 3.2. RFID Technology

In this section, we discuss IoT-based asset tracking systems based on RFID. We focus on RFID first because it is one of the most commonly used technologies. Anandhi et al. propose an authentication scheme for an RFID-based object tracking system [11]. Anandhi et al. criticize tracking systems that use GPS, video cameras, and WI-Fi, and argue that the RFID tags with embedded sensors can provide better tracking. In the report, it is unclear which RFID tags and sensors they used. Moreover, they proposed a communication architecture consisting of four entities, tag, reader, user, and a cloud server. The role of these entities is not described in the protocol; for instance, it is unclear why the user was added to the system, and why the reader cannot authenticate the tags and send the data to the cloud server. In our example system (Figure 2), we clearly defined the functionality of each entity. Our threat model covers the vulnerabilities and attacks for each of them.

Anandhi et al. present a performance evaluation in [11] that is simulation based and does not represent their claim of lightweight operation since the encryption/decryption operations are not executed on the tags. Anandhi et al. criticize existing works on a number of parameters including the difficulty of deployment; however, they do not mention their deployment strategy, so it is not clear how the proposed scheme is better than the critiqued works.

Lie et al. [12] propose an RFID-based asset tracking system for museums. The system uses passive RFID tags and readers to locate objects within a certain distance based on the received signal strength indicator (RSSI) value of the identification signal sent by the tags attached to the art works. Object localization based on RSSI is well established and often used for object tracking; however, the proposed model has several limitations. First, the passive RFID tags and readers can only cover a short range and are not suitable for museums with large collections stored in rooms spread across multiple floors. Second, the movement of assets in museums is portrayed in a rather simplistic way where the objects can only be at a single place storage/display and at a pre-defined distance. If they do not meet these conditions, the event is classified as theft. Third, the RFID tags can not report the condition of the art works. Museums are more concerned about the condition than theft as it is difficult to sell famous pieces of art. Fourth, the passive RFID tags do not support any security and privacy mechanisms as they do not have enough compute and energy resources. Overall, it seems that the model is proposed for object tracking in general rather than designed for the high-value assets in museums. Our proposed threat model is suitable for high-value assets in museums, is more flexible with respect to distance and supports condition monitoring.

Fan et al. [13] propose a cloud-based mutual authentication protocol for IoT devices. The protocol is lightweight as it is based on bit-wise rotation and permutation operations and computationally heavy operations are off loaded to a cloud server. In the proposed protocol, authentication data are encrypted by permutation and updated by the use of a timestamp. However, the reliance on a cloud server for authentication might not always be practical as sometimes IoT devices used in assets tracking scenarios can become disconnected; for instance, when the items are on board a ship or transported in an aircraft. This protocol is not suitable for realistic and practical asset tracking systems. Our reference sensor system uses a reader that can provide authentication and secret key generation services for the piece of art it is attached to.

Masoumeh et al. [14] propose an authentication protocol based on Authenticated Encryption systems. It addresses the security limitations of the SecLAP protocol which was designed for constrained devices. They designed two attacks to analyze the security of SecLAP: first, a passive attack that partially discloses secret parameters and second a full secret disclosure attack that can extract all secret parameters with the complexity of $27n^{7}$. The improved protocol is designed for passive UHF RFID technologies and based on bitwise rotation and XOR operations. The proposed scheme needs to be extended to support encrypted communication of unicast and broadcast messages, authorization, and IAM systems. Additionally, the UHF RFID tags are not equipped with temperature, pressure, accelerometer, and gyroscope sensors to report condition monitoring data rendering this approach not suitable for the tracking of certain high-value assets such as paintings, etc.

Muller et al. [15] propose the use of a distributed ledger (DLT) to monitor the handover management of high-value parcels. Sensors are deployed inside parcels to monitor their contents and log violations of service level agreements. The fact that the sensors are placed inside the packages allows the occurrence of logging during the package handover process (handover from one carrier to another, or from the carrier to the receiver) without the need to open the package, saving time and catching problems early. The sensors in the work are not provided with a physical security system, allowing potentially corrupt data flow from the sensors to the DLT. Our reference architecture includes physical sensor security.

### 3.3. Industry Practices

Now, we discuss IoT-based asset tracking solutions designed and/or used in different industries. We also highlight their limitations, and provide the rationale for them being unsuitable for tracking different types of assets. We discuss tracking solutions that cover transport and on-site use cases. Fortecho is a UK-based asset tracking solution provider. They design monitoring systems for high-value artworks in museums, private collections, and super yachts. The tracker employs active RFID tags equipped with sensors, namely temperature, humidity, pressure, vibration or a three-axis accelerometer to monitor the conditions of artwork. There is a security mechanism to detect whether the tags have been removed. Readers are interrogated at 200 Hz by the backend software to detect whether they have been compromised and/or RF jamming attacks are occurring. Fortecho’s tracking solution has some limitations as discussed below. First, it does not cover art transport meaning that their solutions are tailor-designed for display or storage use cases of super yachts. The dependence of its sensor readers on wired communication (Ethernet or serial) prevents its use for art transport. Second, we noted that the Fortecho sensors perform no edge computation and were limited to only one type of sensor per tag, temperature, humidity or vibration. Such trackers cannot be used in applications with the requirement of sensing multiple conditions simultaneously.

Next, we discuss a few solutions that cover transport. Azure IoT Hub by Microsoft provides a backend to a tracking solution with features to connect, monitor, and manage the IoT assets [2]. It allows bidirectional communication between devices and the cloud with some security features like device authentication and over-the-air updates. Maersk uses Azure IoT Hub to track and monitor the IoT devices deployed on its refrigerated containers as they move around the planet. The condition monitoring data can be monitored so that shipments arrive safely. Recently, Michelin, Argon Consulting and Sigfox France partnered together to create a tracking solution called Safecube that provided real-time supply chain visibility based on condition monitoring [16]. The solution was based on a 0 G network technology with intercontinental coverage in over 60 countries. Another important partnership is among Deutsche Post DHL, ALPS Electric Europe GmbH and Sigfox that aims to design a tracking solution to optimize the individual processes within DHL’s supply chain [4]. The idea is to monitor roll cages with networked sensors to track them in real time with high visibility.

Post Luxembourg provides track and trace solutions for its postal services nationally and internationally. It covers Europe, New York, Hong Kong and Singapore. The tracking solution uses LTE-M tags for the machine to machine (M2M) communication over a proprietary, international LTE network. Hardware specifics for Post Luxembourg’s track and trace service have not been described. The tracking incurs some cost and provides 30 Mb and 50 SMS per month, meaning that at maximum two updates can be sent about the shipment. Thingfox provides asset tracking solutions for trucks, freight trains, cargo ships, and aeroplanes covering different modes of transport such as land, sea, and air. The tracker is of the size of a smart cell phone with a 6000 mAh battery. It is connected to a backend over BLE. It supports global connectivity with LTE and localization is achieved by GPS with Beidou or GLONASS. The Thingfox tracker cannot be used for tracking assets as it is difficult to put a device with such a large battery on many assets such as art as it can affect/damage the condition. Google Cloud also offers asset tracking via LTE and Wifi. Another solution is from Cloud Hawk that offers asset tracking via LTE. LTE is the most commonly used solution, but it becomes expensive for a large amount of communication wherein the tracker must report condition monitoring data very frequently. In all cases, communication security is the same as that provided by cellular communications in general.

### 3.4. IoT in Logistics Systems Security Models and Approaches

There are few studies that investigate the security issues of IoT devices used in logistics. Each of these take a different approach to address the security and are not comparable one with another.

The work in [17] employs a game theoretic approach to model the security the assets in transport. A demon game is modelled between a defender and a quantal response (QR) attacker where the adversary does not have full information of the organization’s network. The proposed solution is a strategy for defending assets in a digital logistics network. A method of successive average-based algorithms is developed to solve the game. The model is evaluated based on numerical analysis for a hypothetical network divided into a few subnetworks, each consisting of a few nodes and a gateway. Such a system is not representative of a real-world logistics system and does not consider practical aspects such as communication that would be the focus of and attack. The evaluation is rather simplistic, and the proposed demon game model might not work for a logistics system with resources constrained devices that cannot run robust security policies.

The work in [18] proposes the use of condition/transportation parameters such as longitudinal, transverse and vertical acceleration to estimate the condition of cargo. Any abrupt changes in the parameters may indicate that damage to the goods has occurred. The parameters are recorded by the navigational communication units installed on trucks. This is an interesting approach to quantify damage and can help in making insurance claims and litigation. The work in [19] proposes to integrate unmanned aerial systems (UAS) to improve the visibility of the assets in a global supply chain. The authors further argue that it is difficult to introduce any new technology into the logistics due to the lack of standards and metrics to evaluate its integration in the legacy systems. In line with this, they gather previously proposed metrics by people working in industry and academia and redefine them to propose the integration of UAS into logistics. Their proposed metrics are defined for a specific case study of a global logistics system for a multinational power systems vendor and cover supply chain performance, material status, warehouse management, business impact, sustainability, and technology adequacy. This is an interesting piece of work and can help the practitioners in the validation and verification of UAS integration into the logistics.

### 3.5. Extant IoT for Logistics Threat Models and Studies

In this section, we review current, published threat models for Wireless Sensor Networks (WSN) and Internet of Things (IoT) systems. We show that there are no application-specific threat models that focus on the communication of IoT devices used for logistics, or address non-malicious threats that may disrupt the operation of the sensor system. We also discuss the threat categorizations and threat model methodologies to justify the categories chosen for our threat model.

The authors of [20] provide a high-level overview of general IoT threats. This paper provides a lot of detail but does not relate the detail to an application. IoT devices, and embedded devices in general, have operational environments defined by their applications. These environments may add new threat vectors. We consider this in our work.

The threat model presented in [21] follows a common pattern in the discussion of WSN/IoT threat classification, reducing attacks into two classes, active and passive. We do not see this categorization lend anything to the understanding of potential attacks, nor their countermeasures, so we do not adopt this approach.

Turakulovich et al. [22] describe common communication layer attacks and specific security protocols for different communication technologies used in WSN. The paper compares the energy consumption of protocols to the number of mitigated threats. It does not provide suitable details to describe the flexibility of the communication protocols for different use cases nor the possibility of new or exacerbated threats due to the application.

Buntun et al. [23] categorize threats by passive and active attacks. They further divide active attacks into a five-layer version of the OSI communication model. Unfortunately, Buntun et al. do not extend the communication model subdivision to the attacks in the passive category. They discuss countermeasures to different attacks offering either general advice or the names of specific research tools. Once again, there is no coupling between a specific IoT application environment and its specific threats.

Jadhav et al. [24] provide less information or structure than the others reviewed here. The paper is notable as the only work to include a small section on environmental, or non-malicious, threats. We also include a taxonomy of such threats that occur in the logistics environment.

Dewal et al. [25] classify protocols as high-level and application-based and describe defensive measures using a security framework. Mamdouh et al. [26] provide a two-dimensional classification of potential threats based on the orientation of the attacks, active or passive, and the communication layer. This paper then offers an interesting introduction to the use of machine learning (ML) for security. We suggest an ML technique, anomaly detection, as a potential countermeasure given our application.

The work presented by Patel et al. [27] is notable because it defines different types of IoT system networks and the different applications that would use these networks. It does not deeply examine the security considerations of each type of network nor applications, but does try and classify the different types of applications. We create a threat model that specifically targets application.

The work presented by Raja et al. [20] outlines threats and attacks for a generic five-layered IoT architecture consisting of application, middleware, internet, access gateway, and edge technology. It further discusses IoT wireless communication technologies, operating systems, communication model, security requirements, etc. This work also highlights some application-based weaknesses, namely spoofing, repudiation, tampering, information disclosure, and DoS present in the OWASP framework. Additionally, some high-level attack surfaces for IoT are described with examples taken from different application use cases. The main issue with this threat model is that when it comes to a specific application use case, the system owners may not know the impact of these attacks, and the system entities that require most protection. Vulnerable system entities may change from one use case to another. For instance, the resources needed for designing a DoS attack on an edge device are different from the one designed for a sensor node. With different types of sensor and IoT devices being deployed in logistics, it is essential that customized threat models are designed for them.

Rizvi et al. [28] highlight the attack surfaces for a user-centric IoT network. The network considered in this work is similar to a distributed computing environment where local users are feeding data into remote servers to be used by data analytics services. The threat model proposed in this work is not appropriate for IoT systems where machine-to-machine (M2M) communication between entities is the primary mode of communication, such as logistics, supply chains, smart industries, etc.

Anand et al. [29] present a generic threat model for IoT systems. A notable contribution of this work is to map threats to attack surfaces and the vulnerabilities that have been exploited to design them. The work further presents the threats and vulnerabilities for two case studies of smart transportation and secure energy management systems. Rizvi et al. [30] present a threat model for IoT devices deployed in healthcare, commerce, and homes. They pick up one or two devices in each domain, underline the threats and compute their vulnerability scores based on the NIST CVSS [31] model. The scores are assigned based on the authors’ understanding and judgement of the likelihood of certain vulnerability to be exploited and the associated attack taking place. We believe it would have been better if the scores were computed from experiments. Our threat model is different from those of Anand et al. [29] and Rizvi et al. [30] because we mapped threats to a superset of the TCP/IP communication model. We believe this mapping is important to identify the security threats for resource constrained IoT devices and take appropriate countermeasures.

Wang et al. [32] present a threat model for trigger-action IoT deployments. The authors argue that the system behaviors in these systems are modelled with complex rules that make it difficult to diagnose the faults and errors. They propose a methodology to infer trigger action rules using Natural Language Processing. This approach is novel but we believe this work is in its early stages and needs time to mature to be used for threat modeling of IoT systems.

Simonjan et al. [33] present a threat model for visual sensor networks security attacks. The threat model is classified based on the STRIDE taxonomy, and security vulnerabilities are mapped to a common weakness enumeration list.

Threat taxonomies such as STRIDE, PASTA, DREAD, and OCTAVE cover the threats for general distributed systems [34].

The main challenge with threat models that are based on the above-mentioned classifications is that they are not suitable for IoT systems with M2M communication.

Anand et al. [35] take an interesting approach to threat modeling based on a machine learning methodology called Transfer Learning (TL). They argue that new threats with varying distributions emerge in different domains from time to time and signature-based anomaly detection algorithms are unable to detect these due to unavailability of labeled data. The learning-based threat model outlines the threats and attacks for a smart home. The authors also show that their proposed model detects unknown threats better than the state-of-the-art models.

The literature on existing IoT-based logistics solutions shows that IoT devices are being integrated into current logistics systems, and that no one technology or approach is suitable for all types of assets.

The most significant limitations are the lack of a security approach to the protection of logistics IoT systems against cyber attacks and the failure to address physical attacks and disruptive non-malicious threats. All of these potential threats need to be addressed in the design and implementation of a logistics sensor system that see use. The cyber security challenges related to logistics and supply chain have not been well studied by research or industry. In this section, we presented current security issues with the use of IoT/WSN systems in logistics.

From the literature review of the existing threat models, it is clear that there is a gap for application-specific threat modeling that includes the most obvious threats such as communication as well as other potential physical and environmental threats. To fill this gap, we propose a threat model designed for logistics and supply chains. In the next section, we present our system model, a generalization of the architecture presented in Section 2; the organization of our threat model, including how we classify threats; and the categories that we include.

## 4. System Architecture and Security Threat Model for High-Value Assets

In this section, we describe the elements and organization of our threat model. We begin by describing a general system architecture for a high-value asset logistics system based on IoT sensor devices that communicate using low-powered radio transceivers. We then describe the elements of our threat model that focuses on the security vulnerabilities associated with the use of small embedded sensing devices that use radio communication.

### 4.1. Example System Model

To better illustrate our concepts, we describe a general asset monitoring or tracking system composed of wireless embedded sensors and processors which we refer to as sensor tags. For illustrative purposes, sensor tags may use 32-bit microcontrollers such as the ARM Cortex M (Microcontroller) class processor [36]. These are larger than 16-bit microcontrollers such as ATmega328 [37] found on Arduino [38], but smaller than the ARM A (Application) class processors found on Raspberry Pi [39] or a typical smart phone processor. A device in this class would have a 64 MHz 32-bit CPU with a floating bit unit, 1 MB flash, 256 KB RAM, and the ARM Crypto Cell [36]. The existence of the ARM Crypto Cell enables some cryptographic processing efficiently on the sensor device, but it does not in itself mitigate the threats described in this paper. The ARM Cortex M class of microcontrollers is a suitable example platform for a logistics IoT sensor node because of its low price, small size and low power consumption.

Our general IoT-enabled asset monitoring or tracking system uses multiple sensor tags. The tags are attached to an asset where possible. Attachment to the asset ensures that the data read by the sensors are those of the asset, and this is performed in a non-destructive way. Where assets are too small, we assume a sensor per asset and we assume that multiple assets are packed together in a way that correlates their sensor data.

Multiple sensor tags attached to the same asset or grouped together in the same container intercommunicate and form a system. A system shares at least one reader node that may have more computing and memory resources than the other sensor tags. Sensor tag to sensor tag, and sensor tag to reader communication is conducted using a low-power radio communication technology like 802.15.4 [40] or Bluetooth Low Energy (BLE) [41]. The reader node has a special radio to communicate the system information, the sensor tags to a backend and the end user. We envision the use of LoRa [42], LTE-M [43], or SigFox [44] communication technologies for this link. Multiple sensor tags that form a system implement resilience through redundancy. Our general, example architecture is illustrated in Figure 2.

Sensor tags periodically broadcast sensor data. The reader and other sensor tags in the same system receive the broadcast. The data are forwarded to the backend by the reader, and can be used by the neighbors to monitor the status of the other nodes in the system, or validate their own sensor readings. We include a super-tag in our example architecture. The super-tag is equivalent to a sensor tag, it senses and broadcasts its data to the reader. The difference is that the super-tag has an extra radio that allows it communication with the backend in the event that the reader becomes untrusted by the sensor tags. In our example system, we assume that the super-tag is indistinguishable from the other sensor tags, including batter capacity. This limitation prevents it from replacing the role of the reader.

### 4.2. Threat Model Elements

We carry out a systematic study to identify potential cyber threats and vulnerabilities that can put the monitoring or tracking assets at risk of compromise. We focus on each attack by providing a definition of the attack and listing the security goals that it violates, the attack type, the threat category as well as countermeasures and additional considerations. Following this, we discuss some threat model elements that enable us to obtain better insights into the threat model.

#### 4.2.1. IoT Trust Model—System Component Categories

Our IoT trust model describes the likelihood of a system component being attacked based on four threat categories, (1) trusted, (2) semi-trusted, (3) trusted but curious, and (4) distrusted. In our example system, each entity is categorized either as trusted or trusted but curious. Sensor tags, super-tags, and the reader are considered as trusted. However, they can be compromised and subject to several cyber and physical attacks while operating in the wild. It is worth noting that the sensor tags and the leader can be compromised, they can malfunction, or become detached from assets during loading/unloading events. Thus, they require continuous behavior monitoring by an anomaly detection system to check whether there has been any change in their performance because, if they are compromised, they can corrupt the system. For the same reason, they need lightweight encryption and authentication security implemented on them to ensure the messages they are sending and receiving are confidential and have not been modified during transmission. The backend is categorized as trusted but curious as it follows the protocol specification in general but gathers information about assets, sensor tags, readers, users, services, and location.

#### 4.2.2. Attacker Model

Our threat model assumes a Dolev–Yao threat model [45] where malicious agents can overhear and intercept messages exchanged between sensor tags, reader, and the backend. We extend the ARM Asset tracking threat model and Security Analysis (TMSA) ARM-PSA report [46] to ensure that all potential attacks are considered. The TMSA covers only four types of attacks, namely impersonation, man in the middle (MitM), tamper, and firmware abuse, while ours is more exhaustive and considers attacks on all layers of a WSN layered architecture.

#### 4.2.3. Threat Categories

We first divide our threats into two categories, malicious and non-malicious, as shown in Figure 3. Our choice of threats is based on those presented in the TMSA [46] but greatly expanded. We aim for our selection of threats to be as complete as possible. Malicious threats are attacks performed by a malicious actor for a purpose. An example of a malicious threat is a Denial of Service (DOS) attack implemented by a malicious agent using a high-power radio transmitter to disrupt sensor communication. Non-malicious threats are risks from the environment that are not performed by a malicious actor but may still affect the operation of the system. An example of a non-malicious threat is the temperature of the sensor and asset increasing during shipping to the point where the sensor ceases to function correctly.

#### 4.2.4. Network Layer Subdivision

We further subdivide our taxonomy of malicious threats by a subset of five layers of the Open Systems Intercommunication (OSI) model (or a superset of the TCP/IP communication model) that is appropriate for embedded systems communication [8]. We arrange malicious attacks by physical, data, network, transport and application layers. It was important that we add the distinction between the physical and data layers found in the OSI model, but remove the session and presentation layers of the OSI model; they relate to web applications more than networks of sensors.

#### 4.2.5. Security Goals

Confidentiality, integrity, availability, (CIA) are widely recognized as three major security goals or high-level properties required by any computer system that handles information or provides a service [47]. Authentication and authorization measures are employed to address these goals and ensure only legitimate entities have access to the system. Next, we define the above security goals in the context of asset tracking. Confidentiality is defined to be the inability of anyone but the system owners to read the data. For an asset tracking system, confidentiality ensures that only the authorized entities (i.e., tags, reader, gateway, and users) can access the secret data such as sensor measurements, keys, credentials, and system logs. Integrity is the correctness of the data generated by any system component. This property guarantees that the data in the system are produced by a valid sensor node and that no data produced by a valid sensor node are modified during storage or transit. Maintaining this property prevents adversaries from injecting fabricated data into the system. Availability guarantees that all entities are available to deliver the services they are designed for at any point in time. This is the property violated by a Denial of Service (DoS) attack aimed at preventing system owners from accessing data or making system changes. CIA also refers to sensor tag properties. Integrity can also be our trust that a sensor node is ours and operating our software, not a sensor tag that belongs to a malicious agent, or has been reprogrammed by a malicious agent to inject false data. Availability can also be a sensor tag level property where the sensor tags are available to send data, and not damaged by a malicious agent, or the environment. We use the CIA properties in our threat model to describe the sensor tag and system properties violated by each attack [48].

#### 4.2.6. Types of Threats

We categorize attacks based on four threat types, namely interception, interruption, modification, and fabrication. This threat categorization loosely describes the attack approach used by attackers exploiting the weaknesses of system entities. A categorization based on attack approach makes this model useful for the design and analysis of countermeasures. For instance, an attack based on interception violates the confidentiality property, and can be implemented with a radio device that can overhear and receive the communication of the sensor nodes. This form of attack may be difficult to detect due to its passive nature. A suitable countermeasure would be communication encryption [45]. An interruption-type attack is easier to detect. An example would be a white noise radio-based Denial of Service attack that violates the availability property and appear as lost data or a faulty system component to the owner. A communication medium designed to be resilient to interference by using channel hopping or other spread spectrum methods helps to maintain system availability [41,42].

A modification attack can be used to violate the integrity property. A malicious agent may capture and reprogram a sensor node to send incorrect data to the system owner. This is a subtle attack that may be difficult to detect. Fabrication-type attacks are similar to modification attacks except that a malicious agent uses a custom-built device instead of a captured one. Attacks based on a fabricated sensor node may be more sophisticated than the ones using a sensor node. The resources of the fabricated node may allow for protocol imitation and subversion, or could perform combined attacks such as spoofing and jamming at scheduled times to maximize network detection while minimizing attack detect ability [49]. Using encrypted keys stored on protected memory on the sensor nodes may help to defend against these sorts of attacks [36].

#### 4.2.7. Threat Countermeasures

We provide suggested potential countermeasures to each attack. Please note that we limit our countermeasures to either data encryption [50] or anomaly detection [51] because these are general, lightweight approaches currently available on low-resourced WSN devices. We leave proofs or formal analysis of the efficacy or correctness of the countermeasures to further work. Indeed, the choice of formalism or methodology is a work unto itself [52]. Our discussion does not preclude the existence of other approaches; the focus of this work is on the categorization of communication vulnerabilities to enable systematic analysis. In the next section, we describe each attack by offering its definition, potential countermeasures, and any additional considerations.

## 5. Threat Descriptions

Each attack in the threat model is described using an attack ID, the sensor device (tag) layer at which the attack takes place, a threat which is an action/event exploiting some existing vulnerabilities to impact the system, the mapping with one of the threats from the ARM threat model, the security goal violation and its impact on the system, and an indicator where the system is vulnerable or is able to detect a threat.

Our threat model is very comprehensive and due to space limitations it cannot be added here. Therefore, for readability, we list the key aspects of the threat model in Table 1 and Table 2. We list assets, attacks and their descriptions, countermeasures and how they can be detected/prevented.

In this section, we discuss the threats in detail following the network layer subdivision described in the previous section. We start at the bottom with the physical layer and then proceed through the data, network, transport, and application layers. For a detailed threat model, please refer to Table A1, Table A2, Table A3 and Table A4. Table 1 describes the attack ID, the asset under threat, the OSI layer, the attack title, and the threat type. Table A2 describes the attack ID, the attack name, the attack definition, and the security property being violated by each attack. Table A3 and Table A4 describe the countermeasure and security consideration that can be implemented to mitigate the effect of the attack. Table 3 describes the non-malicious attacks and their definition for our asset tracking system.

### 5.1. Physical Layer Attacks

The physical layer is the first layer in the WSN protocol stack; it handles the actual interaction with the hardware, defines signalizing mechanisms, and sends and receives RF transmission. The broadcast nature of wireless communication makes it susceptible to jamming, eavesdropping, node tampering, and hardware hacking. On the physical layer, the attacker can target both the tags and reader.

#### 5.1.1. Eavesdropping (**A1**)

The first attack we discuss is eavesdropping, a passive attack where the adversary overhears the broadcast communication between the sensor tags and the reader to obtain insights about the network and condition monitoring data. This information can later be used to carry out active attacks. It is a violation of the confidentiality and integrity goals. Open and distributed systems like asset tracking are vulnerable to eavesdropping, and it is hard to detect such attacks.

#### 5.1.2. Passive Interference (**A2**)

Wireless communication is quite vulnerable to interference which can come from the environment due to obstacles, walls, large-scale path loss fading, and short-scale multi-path fading. Additionally, radio signals become impaired/blocked, or frequency de-tuning can occur due to interference from equipment such as noisy electronic generators, power switching supplies, metal compounds, water or ferrite beads. Such unintentional interference can temporarily disable the tags or disrupt the communication because the radio signal becomes weakened and never reaches the reader, compromising system availability.

#### 5.1.3. Node Tampering

In an asset tracking system, sensor tags and reader are vulnerable to physical node tempering as they can drop off from the asset or be forcefully removed and stolen as discussed below. Attack IDs **A3** to **A6** present such attacks in Table 1 and Table 2. Sensor tags can be permanently disabled by destroying or modifying them. An adversary can remove a tag from the asset, and replace it with another similar tag under its control. An adversary can cut off, crush or puncture the antenna to disable the radio transmission and disrupt data communication. The sensor tag can also be disabled/damaged by placing a Faraday cage around it, such as a microwave oven. Besides tags, a reader can also be stolen, destroyed, or modified if it is deployed in an unattended or unprotected place.

#### 5.1.4. Man in the Middle (MitM)

Another key attack that can occur at the physical layer is the MitM, attack IDs **A7**, **A8**, and **A9** in Table 1 and Table 2. It is a very common attack in wireless communication where an adversary places a malicious device between a tag and a reader to intercept and modify the radio signals. A more sophisticated MitM attack can be designed by making use of multiple devices. There are two types of MitM attacks, (1) mafia fraud and (2) terrorist fraud. In mafia fraud, a malicious device is placed between legitimate devices to capture data packets from the sender, and then modified data packets are relayed to the receiver. Terrorist fraud is similar to mafia fraud but involves some level of collusion with a legitimate sensor tag. A legitimate sensor tag cooperates with the malicious relaying party to convince the reader that the dishonest but legitimate tag is close, and the data packets can be relayed via it.

A MitM attack violates the confidentiality and integrity of the asset tracking systems where an unauthorized entity can gain access to sensitive conditional monitoring data and modify it to change system behavior. For instance, false positive events could be triggered to indicate loading and unloading of assets, a sensor tags battery dying, or the theft of an asset. Relay attacks, like MitM, can be launched from a distance. A sensor tag that was removed from an asset can still communicate with the reader, providing a false impression of it being attached to the asset. As listed in Table 2, these attacks can be detected at the system level by implementing an authentication scheme that periodically pings the tags to ensure they are alive and responding correctly.

Attacks on the physical layer can be detected and/or prevented by employing anomaly detection techniques based on machine learning algorithms to classify benign and compromised node behaviors.

### 5.2. Data Link Layer Attacks

Next, we discuss the attacks (refer to **A10** and **A11**) at the data link layer. This layer supports efficient access to a shared medium to control data transmission and handle any transmission errors that might occur. As the wireless transmission is susceptible to environmental noise and the other physical attacks as explained above, the main aim of the adversary is to carry out attacks that consume network bandwidth and cause packet retransmissions such that other legitimate nodes cannot transmit in their allocated slot. For example, a DoS attack can be launched by jamming the channel and cause sensor tag energy drain by causing constant message back off and packet retransmissions. Asset tracking systems are generally equipped with inertial measurement unit (IMU) sensors, accelerometers, gyroscopes, and magnetometers to measure the force, angular rate, and the orientation of the assets. IMU measurements are prone to drift over time. An intelligent adversary can design attacks to purposefully change the value of IMU sensors in order to generate incorrect data or false positives indicating asset damage or unplanned loading/unloading events. Such attacks compromise the confidentiality and integrity of the asset tracking systems, and often it is difficult to distinguish between these and lossy channel conditions.

### 5.3. Network Layer Attacks

The Network layer is the third layer on the WSN layered stack; it is responsible for packet transmission/forwarding between a sender and a receiver. It also supports routing in a multi-hop network, where intermediate nodes handle data packet forwarding. In such cases, an attacker can carry out various attacks (spoofing/replaying information, selective forwarding, black hole, sinkhole, node replication attack, wormhole, and hello flood). Our proposed architecture does not involve multi-hop transmission so is not vulnerable to attacks aimed at multi-hop routing. Sensor tags, readers, and their protocols are vulnerable to forging, cloning, impersonation, spoofing, replay, and eavesdropping.

#### 5.3.1. Forging, Cloning, and Impersonation (**A12**)

When it comes to sensor tags, there are several different types of sensor tags and RFID tags to choose from. Sensor tags are generally re-writeable and re-programmable, and can easily be replicated. A malicious tag could be placed on an asset to be identified by the reader and subsequently used to trace/locate the artefacts. It is noted that both the sensor tag and reader are vulnerable to impersonation attacks (refer to **A15**). In a sensor tag impersonation attack, an adversary can copy the ID and data of an existing sensor tag to a malicious sensor tag and attach it to an asset to monitor its precise location. A cloned sensor tag can also send fabricated conditional monitoring data to the reader, or be used to initiate other attacks.

#### 5.3.2. Spoofing (**A13**)

A spoofing attack aims to impersonate a sensor tag by capturing a valid ID through eavesdropping and attempts to join the wireless network of the asset tracker. The Spoofing attack requires full access to the communication channel. The reader can also be the target of a spoofing attack where an adversary counterfeits the ID of a legitimate reader in order to elicit sensitive information or modify data on sensor tags.

#### 5.3.3. Replay (**A14**)

The replay attack is where an adversary gains a valid sensor tag ID from an eavesdropping attack and uses it for unauthorized authentication. For instance, a pre-recorded authentication session can be replayed so that the reader believes it is talking to a legitimate sensor tag. The sensor tag could be later removed/replaced, but the reader would still believe that it is still attached to the asset.

#### 5.3.4. Eavesdropping (**A16**)

Eavesdropping on the network layer is initiated by using an antenna that records communication between the sensor tags and reader. The insights gathered from eavesdropping the communication of an asset tracking systems can be used to carry out more sophisticated attacks, such as bypassing an authentication protocol, injecting false conditional monitoring data, or modifying the protocol. A system may not be able to detect this attack because it is passive and it could be very disruptive as it violates system confidentiality and integrity.

#### 5.3.5. Network Protocol Attacks (**A17**)

Last but not least, attackers can exploit the vulnerabilities of the network protocol to modify/change the way entities communicate, broadcast hello packets, hijack the authentication sessions, and escalate the access privileges. These attacks can successfully target sensor tags, readers, and the backend if the protocols are poorly designed and robust security measures are not adopted.

Asset tracking systems are vulnerable to the above attacks; they violate their confidentiality and integrity. Following this, we discuss a few security approaches listed in Table 2 that can mitigate these attacks. Impersonation attacks require authentication and trust methods as countermeasures. Identity and authentication schemes ensure the tags and readers are alive, responding, and have not been compromised, whereas a trust model guarantees they are behaving in an expected manner. The replay attacks can be detected if the system has a robust sensor tag identification scheme and uses timestamps in lightweight encryption (LWE) schemes for freshness. Additionally, LWE schemes, authentication, trust, and periodic key updates can serve as the countermeasures for eavesdropping system network protocols.

### 5.4. Transport Layer Attacks (**A18**)

The transport layer is in control of reliable data transmission between a sender and receiver. The most common attacks on this layer are energy drain, data integrity, and session de-synchronization. For instance, an adversary can hijack the authentication session between tag and reader by modifying the time stamps or replaying old information to prevent the tag from authentication. The asset tracking systems can mitigate these attacks by using a protocol that has robust encryption.

### 5.5. Application Layer Attacks

The application layer abstracts away the underlying physical topology of the network and aggregates data for the asset tracking application. Adversaries target the end-user applications via malware, buffer overflows, and malicious code injections attacks as discussed below.

#### 5.5.1. Unauthorized Tag Reading (**A19**)

This attack can also happen at the application layer where the adversaries attempt to read the contents of sensor tags from a certain close distance, such as less than a meter away. Designing a system level trust model can assist the reader in finding out the discrepancies among conditional monitoring data sent by different sensor tags based on data correlation and acceptable threshold, and later identify the compromised tags.

#### 5.5.2. Tag Modification (**A20**)

Following from previous attack, if the sensor tags are equipped with re-writable memory like SRAM, they can easily be overridden, and the sensitive information can be modified and deleted. Trusted computing or that on node-trusted RAM is one of the ways to protect tags from data modifications [50].

#### 5.5.3. Buffer Overflows (**A21**)

The application middleware can be attacked by exploiting compromised tags to launch buffer overflow attacks on the backend. Unauthorized firmware updates on the sensor tags and reader make asset tracking systems vulnerable to buffer overflows and direct memory access attacks. These attacks can be mitigated by employing trusted computing approaches.

#### 5.5.4. Malicious Code Injection (**A22**)

For this attack, an adversary uses the memory space of sensor tags to store and spread malicious viruses and malware. Mirai bot [53] and Stuxnet [54] are two well-known attacks that were spread by malware and malicious code injections. Stuxnet spread on Microsoft Windows computers via USB sticks as the computers and would have been more catastrophic if the computers were connected to the internet.

The Mirai bot malware targeted resource-limited devices, baby monitors, IP cameras and home routers, and turned them into bots to be used in a large-scale Denial of Service attack (DOS). In a similar fashion, the adversaries can compromise sensor tags and use them to launch a DOS attack on the reader. Moreover, cyber criminals targeting asset tracking systems can employ similar techniques to spread malicious code infecting the applications running on the backend systems. Attacks on the application layer may violate the confidentiality, integrity, availability and authentication of the asset tracking systems and can prove to be catastrophic if proper security measures are not employed.

LWE encryption and authentication schemes can be used to ensure that firmware is upgraded in an authorized way. Moreover, trusted computing techniques can also be used to prevent unauthentic data and middle-ware modifications and upgrades.

### 5.6. Multi-Layer Attacks

Next, we discuss the attacks that can be launched on different layers simultaneously and/or or designed in a way to exploit the vulnerabilities found on more than two layers. Some examples are tracing, DoS, and MitM as listed in Table 1.

#### 5.6.1. Tracing (**A23**)

One of the key requirements of the asset tracking systems is to provide precise and accurate location of high-value assets, thus having robust localization methods deployed therein. There can be instances where the location information is only shared with relevant authorities by employing secure data access and management mechanisms. However, this requirement raises concerns regarding the data privacy as the location information of tags can also be obtained via other data items such as unique tag IDs, thus violating confidentiality. Additionally, system level communication could be used by the attacker to track sensor tags and assets.

#### 5.6.2. DoS and DDoS (**A24**)

Sensor tags and reader both can be targeted and/or compromised to launch a DoS and a distributed DoS (DDOS) attack. A DDoS attack is one where multiple sensor tags work in a distributed way, each performing a DOS attack. When adversaries target sensor tags, the compromised tags can be used to launch a DDoS attack on the reader. Likewise, multiple malicious readers can overload sensor tags with more downlink data than they can handle. These attacks make the asset tracking systems unavailable. One way to secure systems against DoS and DDoS attacks are trust models. Non-compromised nodes can use the trust model to detect the changes in conditional monitoring data and inform the reader. However, the reader can ignore the attack due to local events it is not fully aware of.

#### 5.6.3. Man in the Middle (**A25**)

The MitM attack can be initiated at different layers by intercepting and modifying the messages exchanged between different entities of asset tracking systems.

### 5.7. Non-Malicious Attacks

Next, we discuss the non-malicious attacks that are unintentional, caused by the environment, and originate from physical conditions (noise, humidity, temperature, and shock), humans, animals, or insects as listed in Table 3.

These attacks can damage the asset tracking systems or the asset itself. Following this, we discuss a few of these attacks.

#### 5.7.1. Temperature

Temperature can affect the accuracy of sensors, the oscillation rate of sensor tag clocks, and the transmission power and receive sensitivity of the radio transceiver [55].

#### 5.7.2. Humidity

Humidity can cause corrosion or shorting of components of asset tracking systems and may also affect the asset itself. Sensor packaging needs to be robust to the ingress of humidity and moisture without inhibiting the operation of the sensor and the transceiver. Moisture ingress can be caused by operation in a high-moisture environment or during shipping when moved from a humid environment to a temperature-controlled environment where the moisture suspended in the air condenses upon the sensor components and causes corrosion. The small scale of the MEM components and their connectors make them prone to rapid corrosion [56].

#### 5.7.3. Physical Shock

Physical shock may cause damage to the sensor tags and the assets. Shock can be caused by transportation or routine movement that impacts and causes damage to the sensor tag, its components, or its antenna.

#### 5.7.4. RF Interference

RF interference can cause high communication loss rates. Communication interference can be caused by other radio networks operating in the immediate vicinity, the presence of large electrical motors, microwave ovens, or other devices that produce a large strong electromagnetic field, or if the sensor tags are placed in a vehicle or container that is made of metal and functions as a Faraday cage.

#### 5.7.5. Animals and Insects

Animals and insects can cause non-malicous interference that damage sensor tags. For instance, spiders can build webs in or around the sensor casing, birds can attack blinking LEDs, or curious squirrels can dismantle the sensor tag [55]. All these attacks affect the asset and need to be detected and communicated before the asset is damaged. However, these attacks can be detected but violate the availability and accuracy of asset tracking systems.

## 6. Conclusions

In this paper, we present a threat model for an IoT sensor-based asset monitoring or ta racking system. We loosely define a threat as being anything that may disrupt the operation of the system, either malicious or non-malicious. In conclusion, this work underlies the importance of system security and robustness being designed into an IoT system, regardless of application, from the start. To achieve this, it is important to consider the security threats from an application-specific point of view. This is achieved by providing an enumeration of threats that can be modeled and used for proofs, formal methods, or other rigorous approaches that evaluate IoT logistics security analysis.

We describe a multi-sensor architecture and enumerate communication threats at a subset of the OSI communication layers; we also create a taxonomy-based model that can enable the designers and developers of modern logistics monitoring or tracking systems to be aware of, and design for, the many potential operational and security risks faced by such systems.

## Figures and Tables

**Figure 1 sensors-23-09500-f001:**
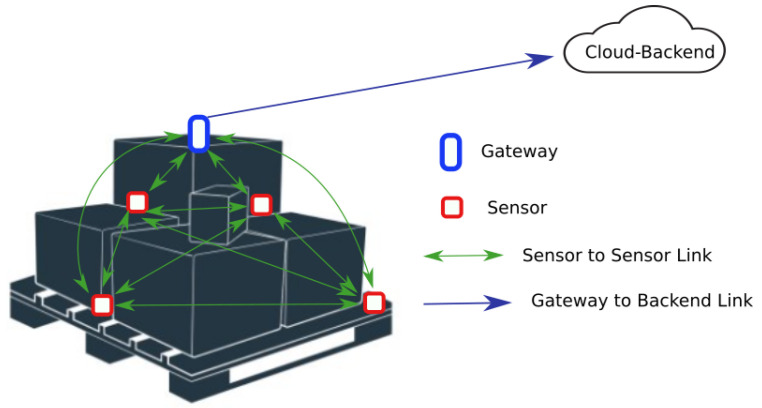
Deployed IoT for the logistics system.

**Figure 2 sensors-23-09500-f002:**
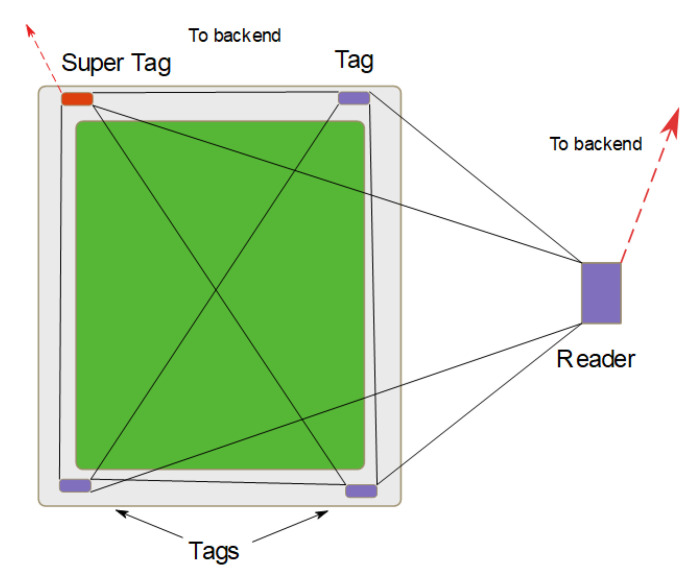
Multi-tag Architecture.

**Figure 3 sensors-23-09500-f003:**
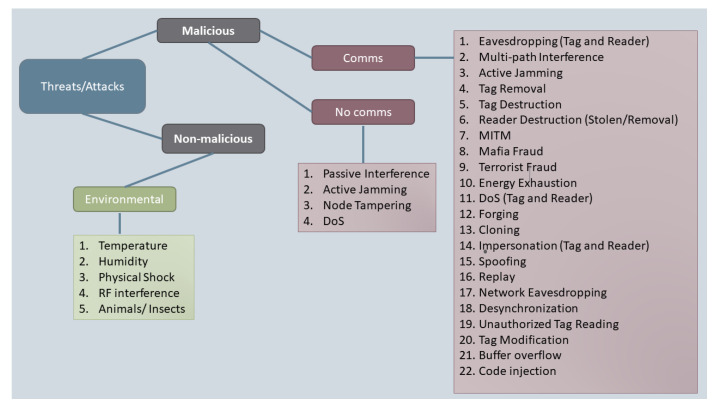
Threat Classification.

**Table 1 sensors-23-09500-t001:** Assets, Attacks, and Definitions.

ID	Asset	OSI Layer	Attack	Definition
**A1**	**Tag**	**Physical**	Eavesdropping	Capture Communication between tag and reader
**A2**	**Tag**	**Physical**	Unintentional or Multi-path Interference	Interference from any source of radio or obstacles
**A3**	**Tag**	**Physical**	Active Jamming (DoS)	Use of high-power radio waves to disrupt communication
**A4**	**Tag**	**Physical**	Tag Removal	Removing, reprogramming and replacing the tag
**A5**	**Tag**	**Physical**	Tag Destruction	Removing antenna, smashing, removing battery
**A6**	**Reader**	**Physical**	Theft, destruction, and removal	theft of reader
**A7**	**Tag & Reader**	**Physical**	Man in the middle	Intercept, modify and repeat modified radio communication
**A8**	**Tag & Reader**	**Physical**	Mafia fraud	Capture and relay information between legitimate devices
**A9**	**Tag & Reader**	**Physical**	Terrorist Fraud	Malicious relaying party tricks legitimate tag to influence the reader
**A10**	**Tag**	**Data Link Layer**	Energy Exhaustion	Subvert communication protocols or sensors to operate inefficiently
**A11**	**Tag**	**Data Link Layer**	DoS	DoS through jamming and interference
**A12**	**Tag**	**Network**	Forging, Cloning, Impersonation	Copy a sensor with all information
**A13**	**Tag**	**Network**	Spoofing	Create malicious device that copies the ID of a trusted device
**A14**	**Tag**	**Network**	Replay	Intercept and repeat authentication information to the reader
**A15**	**Reader**	**Network**	Impersonation	Impersonate a legitimate reader
**A16**	**Reader**	**Network**	Reader Eavesdropping	Record communication between tag and reader
**A17**	**Reader**	**Network**	Network Eavesdropping	Use sensor network protocol to compromise the backend
**A18**	**Tag and Reader**	**Transport**	Desynchronization	Hijack authentication session between tag and reader
**A19**	**Tag**	**Application**	Unauthorized Tag Reading	Read the contents of the tag from a distance
**A20**	**Tag**	**Application**	Tag Modification	Capture sensitive information or modify protocol behavior
**A21**	**Tag**	**Application**	Buffer Overflows	Tags used to perform buffer overflow attack on backend
**A22**	**Tag**	**Application**	Malicious Code Injection	Use tags to infect backend with virus or malware
**A23**	**Tag**	**(Physical, Application)**	Tracing	Unique tag indentification used to find tag location
**A24**	**Tag**	**(Physical, Application)**	Denial of Service (DoS)	Malicious readers overload tags with communication
**A25**	**Tag and Reader**	**(Physical, Application)**	Man in the Middle	Message Interception

**Table 2 sensors-23-09500-t002:** Attacks, Countermeasures and System Considerations.

ID	Countermeasures	System Considerations
**A1**	Anomaly detection and Encryption	Requires Encryption, passive attack, no detection
**A2**	Anomaly detection	System can detect erratic communication
**A3**	Anomaly detection	System can detect erratic communication
**A4**	Anomaly detection and Encryption	System can detect absence
**A5**	Anomaly detection and Encryption	System can detect absence
**A6**	Anomaly detection and Encryption	System can detect, tags need route to communicate with backend
**A7**	Encryption	Assume attacker does not have secret key
**A8**	Encryption	Assume attacker does not have secret key
**A9**	Anomaly detection and Encryption	System can detect anomalous behavior
**A10**	Anomaly detection and Encryption	System can detect anomalous behavior
**A11**	Anomaly detection	System can detect anomalous behavior
**A12**	Anomaly detection and Encryption	System can detect anomalous behavior
**A13**	Anomaly detection	System can detect anomalous behavior
**A14**	Anomaly detection	System can detect anomalous behavior
**A15**	Encryption	Assume attacker does not have secret key
**A16**	Encryption	Assume attacker does not have secret key
**A17**	Anomaly detection	System can detect anomalous behavior
**A18**	Anomaly detection and Encryption	System can detect anomalous behavior
**A19**	Encryption	Assume attacker does not have secret key
**A20**	Anomaly detection	System can detect anomalous behavior
**A21**	Anomaly detection and Encryption	System can detect anomalous behavior
**A22**	Anomaly detection and Encryption	System can detect anomalous behavior
**A23**	None	None
**A24**	Anomaly detection and Encryption	System can detect anomalous behavior
**A25**	AEncryption	Assume attacker does not have secret key

**Table 3 sensors-23-09500-t003:** Non-malicious Threats.

ID	Attack	Definition
**E1**	**Temperature**	Accuracy of sensors, accuracy of clocks, transmission power and sensitivity of the radio transceiver.
**E2**	**Humidity**	MEMs components and their connectors are prone to rapid corrosion and shorts.
**E3**	**Physical Shock**	Physical damage to the sensor node.
**E4**	**RF Interference**	High communication loss rates. Communication interference can be caused by other radio networks.
**E5**	**Animal, Insects**	Non-malicious interference from insects or curious animals

## Data Availability

Data are contained within the article.

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
