# Peer review of "Threat Modeling for Communication Security of IoT-Enabled Digital Logistics"

_sensors, 2023, doi:10.3390/s23239500_

Round 1
Reviewer 1 Report
Comments and Suggestions for Authors
The authors have to address all of the below concerns carefully.
- Abstract: "we" is used a lot in the abstract which makes it poor writing as a scientific research. The abstract requires significant improvement to be clearer and more comprehensive.
- What is "DHL"?
- In contributions, the second point should be combined with the first point.
- The authors have given details about the threat model. We recommend adding some details about the IoT trust model.
- The fourth point is not necessary as it should be present in research such as this one.
- In various places of research, instead of using the word “authors”, the names of researchers should be used.
- Why is the focus on “RFID Technology” as one of the “Threat Considerations Specific to the Use of IoT in Logistics” topic titles? As there are a lot of technologies used in IoT like WBAN, etc.
- Some recent research that addresses threat and trust models that could be useful as references for this research.
o https://www.mdpi.com/2076-3417/10/6/2007
o https://link.springer.com/article/10.1007/s11277-023-10668-x
- What is the purpose of this section "2.6. Discussion"? Also, it doesn't have enough discussion.
- We believe that Figure 2 does not contain sufficient classifications. Why were only these attacks identified? Authors should provide a clear reason.
- Is it possible to provide mathematical expressions or analyzes with proofs to support countermeasures?
- The conclusion of this study is not present (unclear) in the conclusion section.
- Figures: Figure 1 requires a zoom-out. Authors should standardize word call Figures in-text such as "fig" page 3, "figure" page 8. Figure 2 is not used in the text.
- Tables: Table titles should be at the top of the table, not at the bottom. The format of table numbers should follow the style of the journal.
- References list: References should follow the MDPI-Sensors style. For instance, Journal names should be italicized. Some references are not written correctly such as References [2], [5], [7], etc. Some references do not contain enough information such as References [8], [16], etc. Some search names in the reference list begin an uppercase letter for each word (such as [22] ... etc.) and others use only an uppercase letter in the first word (such as [1] … etc.), author should standardize style. The "internet" word must start with a capital letter, see Reference [26]. It is recommended that authors use a standard format for all references. The list of references requires extensive scrutiny.
Comments on the Quality of English LanguageEnglish Writing: This research requires extensive scrutiny of English writing. There are some of grammatical, spelling and typos problems. Authors should use either USA or UK standard English writing (such as "modernise", "analyse", "categorise", etc.). The authors have to thoroughly scrutinize the paper.
Author Response
October 5, 2023
Editor
Sensors Editor
RE: Threat Modelling for Communication Security of IoT enabled Digital Logistics,
Dear Editor,
The authors would like to thank the reviewers for their time and providing insightful comments and suggestions for our manuscript. Please find below a point-by-point response to the reviewer’s comments.
Please note that due to the profound and deep-reaching suggestions by the reviewers, sections have been modified, and added.
- The abstract has been altered.
- The introduction has been reorganised and re-focused.
- A new section called Motivation has been added which illustrates the reason for the refocus of the intro.
- The refocus of the introduction has been highlighted throughout the text where relevant.
These deep changes answer many of the reviewers deeper questions, and are referred to by number in our responses below.
Reviewer 1
The authors have to address all of the below concerns carefully.
- Abstract: ”we” is used a lot in the abstract which makes it poor writing as a scientific research. The abstract requires significant improvement to be clearer and more comprehensive.
We have rewritten the abstract to make it clearer. We, however, maintain the use of the active voice to improve readability.
- What is ”DHL”?
DHL is a major, global shipping organisation, we make this clearer in the text.
- In contributions, the second point should be combined with the first point.
Done, please see text.
- The authors have given details about the threat model. We recommend adding some detailsabout the IoT trust model.
Done, please see text.
- The fourth point is not necessary as it should be present in research such as this one.
Done, please see text.
- In various places of research, instead of using the word “authors”, the names of researchersshould be used.
Done, please see text.
- Why is the focus on “RFID Technology” as one of the “Threat Considerations Specific to theUse of IoT in Logistics” topic titles? As there are a lot of technologies used in IoT like WBAN, etc.
Our literature and state-of-the-art product review showed that RFID tags are the dominant technology used for asset management. We have added this point to the text.
- Some recent research that addresses threat and trust models that could be useful as references for this research.
- https://www.mdpi.com/2076-3417/10/6/2007
- https://link.springer.com/article/10.1007/s11277-023-10668-x
Thank you for the recommending useful references. Reviewer #1 suggested we cite references [1] [2] mentioned above, while we appreciate the reviewer’s suggestion and we could accommodate the request, but, after a careful review of both references, we believe they are not relevant to our research work. We could reconsider these if the Editor and the Reviewer #1 believe they would add value to our manuscript.
- What is the purpose of this section ”2.6. Discussion”? Also, it doesn’t have enough discussion.
This section has been incorporated into the previous.
- We believe that Figure 2 does not contain sufficient classifications. Why were only theseattacks identified? Authors should provide a clear reason.
Done, please see revised Figure.
- Is it possible to provide mathematical expressions or analyzes with proofs to support countermeasures?
We state that we choose our countermeasures based on pre-existing solutions. We also state that we leave formal models and analysis as further work. Indeed, even the selection of an appropriate modelling formalism or model checking tool is a current area of enquiry.
- The conclusion of this study is not present (unclear) in the conclusion section.
The conclusion has been made clearer.
- Figures: Figure 1 requires a zoom-out. Authors should standardize word call Figures in-text such as ”fig” page 3, ”figure” page 8. Figure 2 is not used in the text.
Figure 1 has been Zoomed-out. Figure 2 is now referenced and figure names have been standardised.
- Tables: Table titles should be at the top of the table, not at the bottom. The format oftable numbers should follow the style of the journal
Done, please see text.
- References list: References should follow the MDPI-Sensors style. For instance, Journalnames should be italicized. Some references are not written correctly such as References [2], [5], [7], etc. Some references do not contain enough information such as References [8], [16], etc. Some search names in the reference list begin an uppercase letter for each word (such as [22] ... etc.) and others use only an uppercase letter in the first word (such as [1] . . . etc.), author should standardize style. The ”internet” word must start with a capital letter, see Reference [26]. It is recommended that authors use a standard format for all references. The list of references requires extensive scrutiny.
Latex document changed to use mdpi.bst style for Bibtex references. The bibliography should now conform to the MDPI style requirements.
Comments on the Quality of English Language
English Writing: This research requires extensive scrutiny of English writing. There are some of grammatical, spelling and typos problems. Authors should use either USA or UK standard English writing (such as ”modernise”, ”analyse”, ”categorise”, etc.). The authors have to thoroughly scrutinize the paper.
Done

Reviewer 2 Report
Comments and Suggestions for Authors
This manuscript needs some modifications such as:
1. There is a main problem in the introduction section so, the authors should follow these steps:
It is helpful to analyze the issues to be considered in the ‘Introduction’ section under 3 headings. Firstly, information should be provided about the general topic of the article regarding the current literature which paves the way for the disclosure of the objective of the manuscript. Then the specific subject matter and the issue to be focused on should be dealt with, the problem should be brought forth, and fundamental references related to the topic should be discussed. Finally, our recommendations for solutions should be described, in other words, our aim should be communicated. When these steps are followed in that order, the reader can track the problem and its solution from his/her own perspective under the light of current literature.
2. On what basis were the Attacks, Countermeasures, and System Considerations selected in Table 2?
3. The author should be shown the proposed model to address the research problem.
4. The author should use meta-analysis in the study.
5. Conclusions were displayed poorly. the author should review this section.
6. The references need some modifications and revision.
Comments on the Quality of English LanguageNothing.
Author Response
October 5, 2023
Editor
Sensors Editor
RE: Threat Modelling for Communication Security of IoT enabled Digital Logistics,
Dear Editor,
The authors would like to thank the reviewers for their time and providing insightful comments and suggestions for our manuscript. Please find below a point-by-point response to the reviewer’s comments.
Please note that due to the profound and deep-reaching suggestions by the reviewers, sections have been modified, and added.
- The abstract has been altered.
- The introduction has been reorganised and re-focused.
- A new section called Motivation has been added which illustrates the reason for the refocus of the intro.
- The refocus of the introduction has been highlighted throughout the text where relevant.
These deep changes answer many of the reviewers deeper questions, and are referred to by number in our responses below.
Reviewer 2
This manuscript needs some modifications such as:
- There is a main problem in the introduction section so, the authors should follow these steps:
It is helpful to analyze the issues to be considered in the ‘Introduction’ section under 3 headings. Firstly, information should be provided about the general topic of the article regarding the current literature which paves the way for the disclosure of the objective of the manuscript. Then the specific subject matter and the issue to be focused on should be dealt with, the problem should be brought forth, and fundamental references related to the topic should be discussed. Finally, our recommendations for solutions should be described, in other words, our aim should be communicated. When these steps are followed in that order, the reader can track the problem and its solution from his/her own perspective under the light of current literature.
Please see revised and re-focused introduction.
- On what basis were the Attacks, Countermeasures, and System Considerations selected inTable 2?
Our choice of attacks is mentioned in 4.3.2, our choice of countermeasures is based on technology already available to us 4.2.7, and to meet the listed threats in 4.2.1. System consideration are based on our actual deployment, mentioned in the intro.
- The author should be shown the proposed model to address the research problem.
Our argument is that previous threat models were not complete enough for the design of a practical sensor system, like the one mentioned in our new intro.
- The author should use meta-analysis in the study.
We chose the most up-to-date literature that is currently available.
- Conclusions were displayed poorly. the author should review this section.
This was covered by reviewer 1 question 12, and has been completed.
6. The references need some modifications and revision.
This was covered by reviewer 1 question 15, and has been completed.

Reviewer 3 Report
Comments and Suggestions for Authors
What motivated your research in the context of IoT-enabled digital logistics and communication security?
• Can you explain the significance of securing IoT systems, especially in the logistics and asset tracking domain?
• Could you elaborate on some of the key security concerns or vulnerabilities associated with the integration of IoT devices in logistics and asset tracking?
• How do these security concerns impact the efficiency and reliability of logistics operations?
• You mentioned that existing IoT threat models do not highlight logistics-specific security threats. What are some examples of these logistics-specific threats that your model addresses?
• How do these logistics-specific threats differ from generic IoT security threats?
• Communication security is a significant aspect of IoT systems. What are the primary communication vulnerabilities that you have identified in the context of digital logistics?
• Can you provide examples of potential communication security threats that could impact asset tracking and monitoring?
• Communication security is a significant aspect of IoT systems. What are the primary communication vulnerabilities that you have identified in the context of digital logistics?
• Can you provide examples of potential communication security threats that could impact asset tracking and monitoring?
• Communication security is a significant aspect of IoT systems. What are the primary communication vulnerabilities that you have identified in the context of digital logistics?
• Author can read the following papers to increase the technical strength of the paper:IoT transaction processing through cooperative concurrency control on fog–cloud computing environment,Survey on Service Placement, Provisioning, and Composition for Fog-Based IoT Systems
• Can you provide examples of potential communication security threats that could impact asset tracking and monitoring?
Moderate editing of English language required
Author Response
October 5, 2023
Editor
Sensors Editor
RE: Threat Modelling for Communication Security of IoT enabled Digital Logistics,
Dear Editor,
The authors would like to thank the reviewers for their time and providing insightful comments and suggestions for our manuscript. Please find below a point-by-point response to the reviewer’s comments.
Please note that due to the profound and deep-reaching suggestions by the reviewers, sections have been modified, and added.
- The abstract has been altered.
- The introduction has been reorganised and re-focused.
- A new section called Motivation has been added which illustrates the reason for the refocus of the intro.
- The refocus of the introduction has been highlighted throughout the text where relevant.
These deep changes answer many of the reviewers deeper questions, and are referred to by number in our responses below.
Reviewer 3
- What motivated your research in the context of IoT-enabled digital logistics and communication security?
Our new introduction describes our Singapore deployment, explains the logistics-specific environmental factors that we had to consider, and explains why those are in our logistics threat model.
- Can you explain the significance of securing IoT systems, especially in the logistics and asset tracking domain?
The new introduction discusses that fraud is common in shipping, IoT systems can help combat fraud and ensure quality of the assets during shipping.
- Could you elaborate on some of the key security concerns or vulnerabilities associated withthe integration of IoT devices in logistics and asset tracking?
We mention in the revised introduction that our real world Singapore experience indicated that the environment was difficult and that a threat model had to include all threats to ensure reliability.
- How do these security concerns impact the efficiency and reliability of logistics operations?Reliability is one of the focuses of our threat model.
- You mentioned that existing IoT threat models do not highlight logistics-specific securitythreats. What are some examples of these logistics-specific threats that your model addresses?
Please see the new subsection 2.2 for the answer to this question.
- How do these logistics-specific threats differ from generic IoT security threats?
Please see the new subsection 2.2 for the answer to this question.
- Communication security is a significant aspect of IoT systems. What are the primary communication vulnerabilities that you have identified in the context of digital logistics?
As we try to point out in section 2.2, mobility and lack of access are the application specific issues that exacerbate security issues.
- Can you provide examples of potential communication security threats that could impactasset tracking and monitoring?
We believe that the threat model in general gives example of potential communication security threats. To go deeper we would need to explain the implementation of one of the attacks.
- Author can read the following papers to increase the technical strength of the paper:IoTtransaction processing through cooperative concurrency control on fog–cloud computing environment,Survey on Service Placement, Provisioning, and Composition for Fog-Based IoT Systems
Thank you for the references to the interesting work. Our system model references lowpowered sensor nodes and so sit below the fog layer.
Comments on the Quality of English Language
Moderate editing of English language required.
Kind Regards,
Threat Modelling for Communication Security of IoT enabled Digital Logistics Applicant

Round 2
Reviewer 1 Report
Comments and Suggestions for Authors
The authors did not respond to some comments. Also, some questions were not answered correctly and accurately. The authors have to address all of the below concerns carefully.
- Abstract: This comment still requires a response. It's not well written. "we" is used a lot in the abstract which makes it poor writing as a scientific research. The abstract requires significant improvement to be clearer and more comprehensive.
- What is "DHL"? The authors did not respond to this comment.
- In contributions, the second point does not seem necessary.
- The authors have given details about the threat model. We recommend adding some details about the IoT trust model. The authors response is "Done, please see text." but we did not see their answer in the research.
- The fourth point is not necessary as it should be present in research such as this one. Can you please explain to us which page and which line?
- Why were only these attacks identified in Figure 2? Authors should still provide a clear reason.
- The conclusion of this study is not present (unclear) in the conclusion section. We still believe that the conclusion section is not written correctly.
- Figures: Authors should standardize word call Figures in-text such as "fig" page 4-line166, "figure" page 3-line90.
- Tables: The format of table numbers should follow the style of the journal.
- References list: References should follow the MDPI-Sensors style. For instance, Journal names should be italicized. Some references are not written correctly such as References [2], [5], [7], etc. Some references do not contain enough information such as References [8], etc. Some search names in the reference list begin an uppercase letter for each word (such as [22] ... etc.) and others use only an uppercase letter in the first word (such as [1] … etc.), author should standardize style. The "internet" word must start with a capital letter, see Reference [26]. It is recommended that authors use a standard format for all references. The list of references requires extensive scrutiny.
Comments on the Quality of English Language- English Writing: This research still requires extensive scrutiny of English writing. There are some of grammatical, spelling and typos problems. Authors should use either USA or UK standard English writing (such as "modernise", "analyze", "categorise", "categorized", etc.). The authors have to thoroughly scrutinize the paper.
Author Response
November 7, 2023
Editor
Sensors Editor
RE: Threat Modelling for Communication Security of IoT enabled Digital Logistics,
Dear Editor,
The authors would like to thank the reviewers for their time and providing insightful comments
and suggestions for our manuscript. Please find below a point-by-point response to the reviewer’s
comments.
Please note that due to the profound and deep-reaching suggestions by the reviewers, sections
have been modified, and added.
1. The abstract has been altered.
2. The introduction has been reorganised and re-focused.
3. A new section called Motivation has been added which illustrates the reason for the refocus
of the intro.
4. The refocus of the introduction has been highlighted throughout the text where relevant.
These deep changes answer many of the reviewers deeper questions, and are referred to by
number in our responses below.
Reviewer 1
1. Abstract: This comment still requires a response. It’s not well written. ”we” is used a lot in the
abstract which makes it poor writing as a scientific research. The abstract requires significant
improvement to be clearer and more comprehensive.
The abstract has been altered. Please see the blue coloured text, lines 12-18.
2. What is ”DHL”? The authors did not respond to this comment.
We have mentioned in the introduction that DHL is a major, international global courier,
highlighted in blue, lines 27-28. Unfortunately, the acronym DHL comes from the surnames
of the founders, Larry Hillblom, Adrian Dalsey and Robert Lynn, providing no revealling
information for the reader.
3. In contributions, the second point does not seem necessary.
We have removed the second point in the contribution as recommended.
4. The authors have given details about the threat model. We recommend adding some details
about the IoT trust model. The authors response is ”Done, please see text.” but we did not
see their answer in the research.
We have described the IoT trust model in section 4.2.1 - IoT Trust Model - System Component
Categories and highlighted it in blue font. Lines 464 - 477.
5. The fourth point is not necessary as it should be present in research such as this one. Can you
please explain to us which page and which line?
We are extremely sorry that we did not understand this comment clearly, and we will highly
appreciate it if the reviewer could explain which fourth point in which section of the paper
are you referring to. So, we would be able to address it.
6. Why were only these attacks identified in Figure 2? Authors should still provide a clear
reason.
Our choice of attacks is mentioned in 4.2.3, Lines 488 - 489, our choice of countermeasures
is based on technology already available to us 4.2.7 Lines 553-560, and to meet the listed
threats in 4.2.1. IoT Trust Model - System consideration are based on our actual deployment,
mentioned in the introduction.
7. The conclusion of this study is not present (unclear) in the conclusion section. We still believe
that the conclusion section is not written correctly.
The conclusion has been made clear, is highlighted in blue, and can be found at lines 817-822.
8. Figures: Authors should standardize word call Figures in-text such as ”fig” page 4-line166,
”figure” page 3-line 90.
We have now used a consistent notion of Figure to represent all the figure numbers throughout the paper.
9. Tables: The format of table numbers should follow the style of the journal.
We have now followed the journal style for table numbers.
10. References list: References should follow the MDPI-Sensors style. For instance, Journal names
should be italicized. Some references are not written correctly such as References [2], [5],
[7], etc. Some references do not contain enough information such as References [8], etc.
Some search names in the reference list begin an uppercase letter for each word (such as
[22] ... etc.) and others use only an uppercase letter in the first word (such as [1] . . . etc.),
author should standardize style. The ”internet” word must start with a capital letter, see
Reference [26]. It is recommended that authors use a standard format for all references. The
list of references requires extensive scrutiny.
References fixed by using the mdpi.bst file on overleaf.
